# Efficient learning of molecule properties with Graph neural networks and 3D molecule features

Anonymous Full Paper
Submission 27

## Abstract

Graph Neural Networks (GNNs) have emerged as a powerful tool in predicting molecular properties based on structural data. While GNNs excel in identifying local patterns within molecules, their ability to capture global properties remains limited due to inherent structural challenges such as oversmoothing.

We introduce an innovative GNN-based model that integrates global 3D molecular features with standard graph representations to enhance the prediction of molecular properties. The proposed model is evaluated using benchmark datasets ESOL and FreeSolv and it outperforms existing models. It demonstrates the crucial benefit of giving GNN models easy access to global information about the graph, in the context of applications to chemistry.

Additionally, the model's architecture allows for efficient training with relatively modest computational resources, making it practical for widespread application.

## 1 Introduction

The development of machine learning models for chemistry is an important direction of research. Many efforts have been done in this direction in machine learning in particular for predicting molecule properties from their structure [1]. A particular kind of deep learning models, called Graph Neural Networks show extremely good results in a variety of tasks on molecule datasets [2, 3]. Indeed, the best models as of now are based on GNNs that sees a molecule as a graph with the nodes being the chemical elements and the edges being their bindings. They are able to recognize patterns inside molecules and relate them with molecule properties.

Although GNNs are extremely good at identifying local patterns, they still struggle to identify or evaluate more global properties of molecules. Due to their structure and their aggregation process, GNNs are prone to oversmoothing [4] and have difficulties grasping information from many nodes of a graph at the same time or from nodes far apart. These limitations to identify more global information also explain why approaches using expert-crafted descriptors are still competitive with GNNs in chemistry [5]. However, global properties may be very important for the prediction of some molecule properties and it is crucial that the machine learning model has an easy access to them. In the case of global properties such as 3D shape, there exists a body of work using GNNs [6] to predict them. So, in principle, (special types of) GNNs are able to get this information. However, the GNN models for these kinds of tasks are more complex and computationally heavy. Recent results, e.g. the Uni-Mol model [7],a large model based on transformers, shows that molecule 3D information is of high importance for predicting molecule properties.

In this work, we propose a new machine learning model, based on a GNN, to predict molecule properties. We call it TChemGNN, for Tiny Chemistry Graph Neural Network. The novelty is that we provide global 3D features as additional input to the standard atom properties and graph. This information is derived from chemistry principles and computed from the standard molecule description. We modify the structure of the GNN so that it makes an efficient use of this additional features. This greatly enhance the predictions. We show on different benchmark datasets (ESOL and FreeSolv) that it outperforms actual, much larger, models. Our model is relatively small and can be trained efficiently with small computer resources. This is an important point for applications.

## 2 Previous work

In order to evaluate the efficiency of machine learning models on chemical tasks, several benchmark datasets have been made openly available online by the community. There is even a website "Paperswithcode.com" keeping track of the performance of the different models in the literature. This is very convenient to test new architectures and new concepts, such as the one presented here. We choose the open-source libraries ESOL [8] and FreeSolv [9] as our datasets for predicting molecular properties and demonstrating the advantages of our model. The task of the ESOL dataset is to predict water solubility (log solubility in mol/L) for common small organic molecules, while FreeSolv provides both experimental and calculated hydration free energy data for small molecules in water.

Presently, the best models on the ESOL and FREESolv datasets are: Uni-Mol, A Universal 3D

Molecular Representation Learning Framework [7], ChemRL-GEM, Geometry Enhanced Molecular Representation Learning for Property Prediction [10], and SPMM, the Bidirectional Generation of Structure and Properties via a Single Molecular Foundation Model [11] or for FREESolv, ChemBFN, A Bayesian Flow Network Framework for Chemistry Tasks [12]. Most of the current best models are large models based on the Transformer architecture. They are trained in a self-supervised manner on large datasets. Their latent representations are then used for classification or regression tasks on other, possibly smaller, datasets. This approach is called Molecular Representation Learning (MRL). These models are able to to create their own representation of molecules and perform well in a variety of applications related to chemistry, but they are costly to train.

We can notice a smaller architecture among the best models called MPNN and its variants [5]. It is based on message passing, i.e. a graph machine learning architecture. Before the MRL trend, graph neural nets were commonly (and are still) used to predict chemical properties. In this framework, the standard setting is to create a graph from the molecule with its atoms bindings and add the atoms descriptors as features on the nodes. These models do not use any global or 3D shape information of molecules as input. As pointed out in [5], due to the small size of datasets and the reduced number of message passing layers, the model's focus is more on the local molecular structure and connections between chemical elements (local features). It has difficulty to learn more global features at the scale of the entire molecule. They therefore add to their model some global molecule information at the last layer of their neural network. This is done by concatenating the latent representation with a vector of pre-computed global features. Our approach is somehow similar, but with a different architecture (graph attention layers) and a concatenation of global information directly at the node level, at the input. We also use a highly reduced set of global features (5 instead of 200), focusing on only 3D properties.

Finally, it is important to mention non-deep learning approaches that may be still competitive. Over time, chemists have developed formulas and relationship between the atomic composition of a molecule and its properties. Many of the most important and useful expert-crafted descriptors can be computed using the open source Python library RDKIT [13]. In particular, hundreds of molecule features can be generated from the SMILES the "simplified molecular-input line-entry system" that encode the structure of a molecule. These descriptors are used in [5] and in our model. We even run a random forest regression using them and show that it gives results on par with the largest deep learning models for FREESolv.

# 3 The TChemGNN model

## 3.1 Structure

Our GNN model is depicted on Fig. 1. It consists of 4 layers of Graph attention network "GATConv" with a hyperbolic tangent as their associated nonlinear function. The number of hidden channels is 56 for all layers. The optimizer is RMSprop. The model is relatively small with a total number of learnable parameters around 13K.

An important aspect of our model is that, at the last layer, the output is a single number given by a particular node of the molecule graph. We have noticed that performing a global pooling operation does not give satisfactory results. We hypothesize that the molecule properties we predict depend only on a part of the graph and the rest provide some random noise that pooling is unable to filter. More precisely, we have noticed that the atoms at the periphery of the molecule are the most able to predict the quantities the model is trained for. This is the case at least for the datasets ESOL and FreeSOLV. It turns out that, the encoding rules of the SMILES notation always put in the first position one of these peripheral nodes [14, 15]. Hence by building the graph such that the node with ID 0 correspond to the first atom in the SMILES encoding, we are able to extract the prediction from this position.

## 3.2 Input features

Concerning the feature space, each node of the input graph has 14 features. 9 of them are local (atom) features that describe each chemical element. We add to each node of the graph 5 global features, carefully selected, among the set of molecular descriptors provided by RDKit. By concatenating these 5 features, we allow a direct access to important global information at the node level. We ignore edge attributes in our model.

More precisely, the atom features are: atomic degree, ring structure within the molecule, number of hydrogen atoms, number of bonds in molecule, surface area, formal charge. Note that the last three features are scaled [16]. The atomic mass scaled $A_s$ is given by: $A_s = (A - 10.812)/116.092$, where $A$ is an atomic mass. The Van der Waals radius ($R_{vdw}$) of chemical elements in a molecule is scaled as: $R_{vdw,s} = (R_{vdw} - 1.5)/0.6$. The covalent radius scaled $R_{cov,s}$ is calculated according: $R_{cov,s} = (R_{cov} - 0.64)/0.76$, where $R_{cov}$ is a covalent radius for each chemical element in a molecule. Atomic number, number of valance electron, and hybridization can be applied for adjusting the results. For the global features, computed with RDKit, one

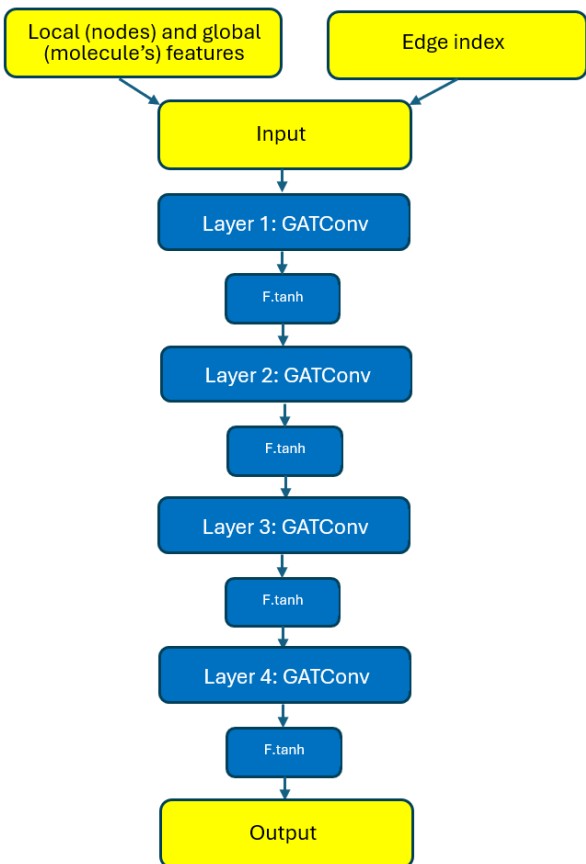

**Figure 1.** The architecture of the TChemGNN model for solubility prediction (ESOL dataset). Both node features and the molecule graph are given as input to the Graph neural network. The feature vector at each node contains the properties of the related chemical element as well as 5 additional 3D global properties (same properties for all nodes). The network is composed of 4 layers of Graph Attention Network and hyperbolic tangent nonlinear functions. There is no global pooling at the end and the output of a single node is evaluated. This output node corresponds to the first atom appearing in the SMILES encoding of the molecule (always a weakly connected node at the periphery).

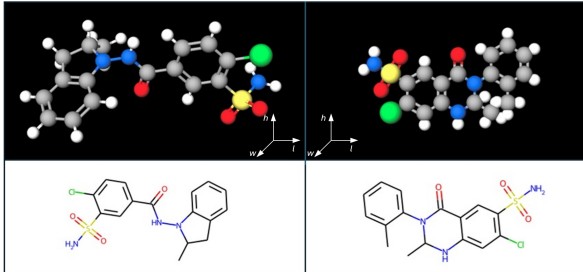

**Figure 2.** 3D representations of molecules with same chemical formula C16H16ClN3O3S from the ESOL library. Even the same chemical formula and a very close structure, these 2 molecules organize differently in space. The graph structure is important, but the shape in 3D as well, for predicting molecules properties. Left molecule (Indapamide): $l = 7.312$, $w = 11.546$, $h = 5.749$ and right molecule (Metolazone): $l = 5.242$, $w = 13.002$, $h = 6.328$.

## 4 Results on ESOL dataset

Our first benchmark dataset is ESOL. The task is to predict the solubility of a molecule. Water solubility is given in log-scaled mols per liter. The dataset contains 1,128 compounds. The measured solubility values range from -8.057 to 1.071.

We list in Table 1 the current best models on this dataset as reported on the website "paperwith-code" [8] and compare to our model.

**Table 1.** Results of our and the state-of-the-Art (SOTA) models for solubility predictions.

| SOTA models for the ESOL library | RMSE |
|---|---|
| Uni-Mol: A Universal 3D Molecular Representation | 0.788 |
| ChemRL-GEM: Geometry Enhanced Molecular representation learning method (GEM) for Chemical Representation Learning (ChemRL) | 0.798 |
| SPMM: Structure-Property Multi Modal foundation model | 0.810 |
| ChemBFN: Bayesian Flow Network framework for Chemistry tasks | 0.884 |
| ChemBERTa-2 (MTR-77M): Masked-language modelling (MLM) and multi-task regression (MTR) | 0.889 |
| D-MPNN: Direct Message Passing Neural Network | 1.050 |
| **TChemGNN (Our model)** | **0.5669** |

Some models are not reported on the website but can have better scores. For example, a model combining long- and short-term memory units (LSTM) with a graph attention network (GAT) has an RMSE of $0.885 \pm 0.067$ [18] and a model called MPNN has an RMSE of $0.700 \pm 0.073$ [19]. This latter model provides the best RMSE score so far to our knowledge. We can also cite the work of [20], with an even better RMSE of 0.569. However, the initial dataset is reduced from 1128 to 1068 molecules. Several molecules that are difficult to classify were filtered out. While this may be relevant for chemists (gases and solids where solubility is not a meaningful prop-

of them is the *dipole momentum*. It involves the charge distribution between chemical elements and their distances and details about molecular interactions. In addition, we add the *angle of general molecular orientation* which is linked to the molecular properties that are strongly influenced by the molecule's shape [17] and orientation in space. Finally, we add three global features, the *width* ($w$), the *height* ($h$) and the *length* ($l$) of the molecule, related to its 3D configuration. To show the importance of these spatial features, we illustrate on Fig. 2 an example of 2 molecules with the same chemical formula. They have the same chemical elements but different spatial organization with a more or less compact shape. This difference causes changes in their behavior and properties.

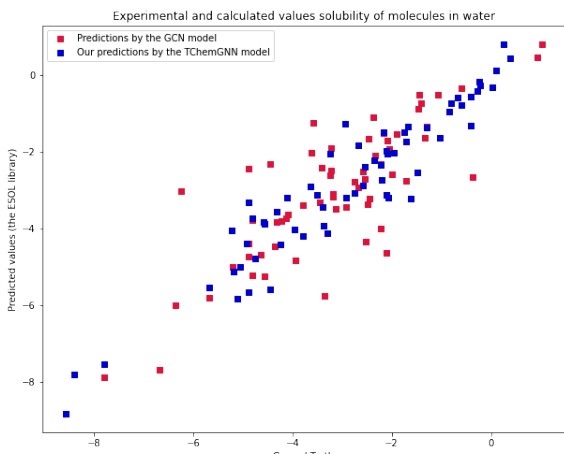

**Figure 3.** Solubility prediction with respect to ground truth (ESOL test dataset). Our model (blue dots) is compared to a basic GCN model (red).

erty) it can not be compared to the other ones. We focus our experiment on the full dataset but we noticed that our results are better than [20] on their subset.

Our model TChemGNN provides the best results for solubility prediction and outperforms all the models (Table 1). The RMSE of our GNN model with global features is 0.5669 on the test set of the Uni-Mol model.

To understand better why our model perform so well, we have made an ablation study. A basic GNN model having four standard GCN layers [21] has been implemented, the "GCN model", to see the impact of GAT layers and global features. We also have modified our network, training it without the 3D features. Note that the hyperparameters of our model (optimizer, step size) are different from the basic GCN model. Results are reported in Table 2. The Graph attention layer and the addition of 3D features both improve the predictions on ESOL dataset.

**Table 2.** Ablation study for our model experimented on the ESOL library.

| model | RMSE |
|---|---|
| GCN model | 1.047 |
| GATConv instead of GCN | 0.7014 |
| Adding 3D features to GCN | 0.7688 |
| Our model without 3D features | 0.7904 |
| **Our model** | **0.5669** |

In order to have a better overview of the prediction results, we show in Fig. 3 the prediction error of TChemGNN for all the molecules in the test set and compare to the GCN model (on the same test set). The error does not seem to depend on the value to predict (the error is even smaller for extreme low and high values).

# 5 Results for the FreeSolv dataset

The FreeSolv dataset is an open database with hydration free energies for a set of 643 neutral molecules, most of which are fragment-like [22]. The data values to predict ranges from -25.47 to 3.43. One of the best models for this dataset is ChemBFN [9, 12]. It is a language model trained on molecule encodings (such as SMILES). The embeddings are then used for classification and regression. It is relatively large with 54M learnable parameters (compared to the 13K of our model).

For this dataset, we first performed a "simple" random forest regression using purely expert-crafted molecule features obtained from [23] and RDKit. This gave an RMSE almost as good as the best deep learning model, ChemBFN. Again, expert-crafted features are extremely powerful for chemistry applications.

We slightly modified TChemGNN from ESOL to FREESolv: one GatConv layer was removed. On Table 3, we show the results of several state-of-the-art models and compare to our model and the random forest experiment. Again TChemGNN outperforms the other models by a large margin. The test and validation datasets were randomly selected. The result on the validation set was RMSE = 0.9003 ± 0.1414 (5 fold cross validation) and on the test set it was RMSE = 1.0342 ± 0.2281. The predictions were quite stable.

# 6 Results discussion

Our GNN model delivers better results on ESOL and FREESolv compared to any other known models. We now analyse why. Several of our results show evidence of the crucial role of expert-crafted features, particularly the global 3D features. Firstly, the ablation study of the models trained on the ESOL dataset show a big difference in the performance when trained with and without the 3D features. Secondly, for the FREEsolv dataset, a random forest ran purely on chemist's features has almost the same results as the best deep network with millions of learnable parameters. These expert-crafted features are underestimated in the machine learning literature. The ultimate proof comes from our small model (̃13K learnable parameters) equipped with a few 3D features that outperforms all the state-of-the-art.

We should emphasize here that the output of our model is not standard. The final output value is taken from a single node rather than performing a global pooling. Any global pooling we tested performed worse than this choice. We assume that, at least for our datasets, the predicted property depend

**Table 3.** RMSE of SOTA models for predictions of hydration free energies of small molecules in water [9].

| SOTA models for the Freesolv library | RMSE |
|---|---|
| ChemBFN: A Bayesian Flow Network Framework for Chemistry Tasks | 1.418 |
| Uni-Mol: A Universal 3D Molecular Representation Learning Framework | 1.620 |
| SPMM: Structure and Properties Through a Single Molecular Foundation Model | 1.859 |
| ChemRL-GEM: Geometry Enhanced Molecular Representation Learning for Property Prediction | 1.877 |
| D-MPNN: Direct Message Passing Neural Network | 2.082 |
| GROVER (base): Self-Supervised Graph Transformer on Large-Scale Molecular Data | 2.176 |
| GROVER (large): Self-Supervised Graph Transformer on Large-Scale Molecular Data | 2.272 |
| N-GramRF: Unsupervised Graph Method | 2.688 |
| PretrainGNN: Pre-training Graph Neural Networks | 2.764 |
| N-GramXGB: Simple Unsupervised Representation for Graphs, with Applications to Molecules | 5.061 |
| Random Forest Regression | 1.4222 |
| **TChemGNN (Our model)** | **1.0342** |

on a combination of 1) global information and 2) particular local patterns inside the molecule. If one of them is missing, the accuracy drops. Hence, the prediction will be correct only around the position of the particular local pattern on the molecule graph. Outside this area, the prediction may be noisy or even random. Global pooling is therefore not suited to this configuration. Then, the important question is: which node to choose for the correct prediction? We found that it is the first node of the graph. Indeed, as discussed earlier, this first node is also, by construction, the first atom in the SMILES encoding. And the encoding rules of SMILES always put first the most peripherical node, with a weak connection to the rest of the molecule. This atom is more prone to interact with other molecules and shape important molecule properties. It seems to be the case for the properties predicted in the ESOL and FREESolv datasets.

In conclusion, our model performs better because we use efficient inductive bias. We make use of knowledge from chemistry, both in the input (3D features) and in the structure of the neural net (choice of the single node output).

## Conclusion

Our study demonstrates significant improvements in predicting molecular properties on two reference benchmarks for chemistry applications of machine learning, ESOL and FREESolv. Our GNN model integrates both chemical element properties and some general properties of molecules, making it an hybrid deep learning architecture enhanced with a few inputted expert-crafted features. These features contain global 3D properties that reflect molecular shape and orientation.

This work highlights the importance of 3D molecule features for the prediction of molecule properties, and the limitation of GNNs to learn them from the molecule graph. It suggests that further modifying of feature selection and model architecture at the interplay of local and global features could lead to even greater predictive accuracy. This approach could be used for other molecule properties and datasets.

Finally, our model has a very modest size and can be trained in a fast manner, even without an access to GPUs, something very convenient for Chemists.

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
