# OpenReview forum: "Efficient learning of molecule properties with Graph Neural Networks and 3D molecule features"
_NLDL.org/2025/Conference — Submitted to NLDL 2025_

### Official Review · Reviewer_o75a · 2024-09-28
**TChemGNN Proposed as New Property Predictor for Molecules**

**Confidence:** 4

**Summary:**

The paper proposes a new GNN architecture called TChemGNN for molecular property modeling. In Section 1, the paper outlines some background related to using GNN for molecular property prediction and in Section 2, the paper discusses some prior work. Section 3 introduces the proposed TChemGNN model, which borrows the architecture from prior work and adds global features to the input representation. Section 4 describes the results on the ESOL dataset and Section 5 describes the results on the FreeSolv dataset. Generally both experiments suggest that TChemGNN achieves lower error values compared to the baselines studied. Section 6 provides a discussion and conclusion that summarizes the main results.

**Strengths:**

* The paper provides many relevant details related to the proposed architecture, the input space and the experiments conducted.
* The experiments presented generally show lower error for the proposed method.

**Weaknesses:**

* The paper does not provide discussion of relevant prior work pertaining to GNNs that use global features for modeling [1] [2].
* The paper does not describe relevant details of the datasets used for the experiments (e.g., their size) that would be relevant for understanding some of the claims.
* The claims about the proposed TChemGNN being fast to train are not supported with evidence and could be confounded with the size of the dataset.
* Since the paper claims improvement in features for representation learning, I would also recommend discussing different molecular representations, such as SELFIES [3], Group SELFIES [4] and SAFE [5].

[1] Chen C, Ye W, Zuo Y, Zheng C, Ong SP. Graph networks as a universal machine learning framework for molecules and crystals. Chemistry of Materials. 2019 Apr 10;31(9):3564-72.

[2] Chen C, Ong SP. A universal graph deep learning interatomic potential for the periodic table. Nature Computational Science. 2022 Nov;2(11):718-28.

[3] Krenn M, Häse F, Nigam A, Friederich P, Aspuru-Guzik A. Self-referencing embedded strings (SELFIES): A 100% robust molecular string representation. Machine Learning: Science and Technology. 2020 Oct 28;1(4):045024.

[4] Cheng AH, Cai A, Miret S, Malkomes G, Phielipp M, Aspuru-Guzik A. Group SELFIES: a robust fragment-based molecular string representation. Digital Discovery. 2023;2(3):748-58.

[5] Noutahi E, Gabellini C, Craig M, Lim JS, Tossou P. Gotta be SAFE: a new framework for molecular design. Digital Discovery. 2024;3(4):796-804.

**Justification:**

While the paper provides a potentially interesting and useful addition for molecular modeling, I think that the paper needs to properly justify the experiments performed (i.e., why those particular datasets were chosen) and provide more evidence for some of their claims (e.g., compute details for "light model training"). On top of that, it would be useful for the authors to perform an ablation with their proposed representation on other model architectures.

---

> ### Author Rebuttal · Authors · 2024-10-19
>
> Thank you for your valuable feedback.
>
> You will find a common global answer to all the reviewers posted as an answer to reviewer #1. Please have a look at it. In addition, we address some of your particular concerns here, in the rebuttal of your review.
>
> Thank you for the suggested references, we already cite a few papers related to the same topic in section 2 but we will add [1] and [2] too. But we don't want to spend too much time discussing the literature on large models and foundation models, we want to go straight to our point due to page limits.
>
> We will add the dataset sizes in the paper.
>
> About molecule representations, we agree that a discussion on references [3,4,5] would be interesting. However, these features, that come from the latent space of large / foundation models, are a bit out of the scope of the paper. But definitely, we plan to consider them in future work. It would be much more convenient from a machine learning point of view to be independent of “handcrafted” features and use other deep net latent representations. This is clearly a good idea.
>
> Thanks again for your valuable insight.

---

### Official Review · Reviewer_q17A · 2024-10-03
**Efficient learning of molecule properties with Graph Neural Networks and 3D molecule features**

**Confidence:** 3

**Summary:**

This work aims to improve performance on two molecular benchmark tasks, FREESOLV and ESOL, using selected global molecular features and a unique single-node-output approach from a graph attention network. The results presented on these two datasets are very strong compared to the included benchmarks.

**Strengths:**

- The results presented in the paper are very strong
- Previous work is well detailed, although more could be said about representation learning with GNNs for molecular properties (GraphCL, InfoGraph, etc.), and the GATConvs used
- Strong detail is given on datasets and features
- The source (paperswithcode) is given for benchmarks is given, which improves my trust in the benchmark selection process
- The single-node-output is unique and qualitatively well motivated

**Weaknesses:**

- The range of datasets used, only two, is limited.
 - The Open Graph Benchmark (OGB) is the usual source for the benchmarks used - but for molecular graph property prediction, it includes many more datasets than used in this work.
 - Several other works have used 3D atom positions for these tasks (see References below), so the claims about novelty could be toned down. The improvement in performance is significant enough without drastic novelty in your method.
 - Error bounds are not produced for main tables, including for benchmark models
 - More detail should be given on hyper-parameters and feature selection
 - Results with pooling outputs should be included alongside benchmarks or with the ablation study
 - The standard scaffold splits for the datasets are not used
 - The style of the paper is closer to that of a masters dissertation than a peer-reviewed publication (for example the network architecture schematic is not needed). I'd recommend rewriting with an eye to providing necessary and useful detail on the experimental methodology and existing works used, for example hyperparameter selection and GATConvs.

**Questions**
These are questions about the whole work, not just the weaknesses above.

- Why only use these two datasets?
- How did you determine which molecular properties to use?
- How did you determine your hyperparameters? What about other training details?
- What is the "most peripheral node"? How is this determined, and how do results differ when using other nodes?

References:

Liu, S., Wang, H., Liu, W., Lasenby, J., Guo, H., and Tang, J., **“Pre-training Molecular Graph Representation with 3D Geometry”**, <i>arXiv e-prints</i>, Art. no. arXiv:2110.07728, 2021. doi:10.48550/arXiv.2110.07728.


*Stärk, H., Beaini, D., Corso, G., Tossou, P., Dallago, C., Günnemann, S. &amp; Lió, P.*. (2022). **3D Infomax improves GNNs for Molecular Property Prediction.** <i>Proceedings of the 39th International Conference on Machine Learning</i>, in <i>Proceedings of Machine Learning Research</i> 162:20479-20502 Available from https://proceedings.mlr.press/v162/stark22a.html.

**Justification:**

This paper presents a large improvement in performance over two molecular benchmark datasets. Several steps are unique and interesting, in particular the single-node readout instead of pooling, which is well motivated. However only two datasets are used, and further, no error bounds are given, so statistic significance cannot be properly attributed. Additionally, despite the claims in the related work of the paper, other works have included 3D global features alongside local features. This means that in the absence of further benchmarks, the paper is essentially "Using a single output node and specific global features improves performance on two datasets". This is a valid contribution, but without further detail on experimental design and error bounds, I cannot recommend publication.

---

> ### Author Rebuttal · Authors · 2024-10-19
>
> Thank you for your valuable feedback. You will find a common global answer to all the reviewers posted as an answer to reviewer #1. Please have a look at it. In addition, we address some of your particular concerns here, in the rebuttal of your review.
>
> The style of the paper is made to be understood also by Chemists with whom we collaborate. The network schematic architecture is there for them. We hope chemists can make use of the neural net we propose for their application even if they don’t have a background in machine learning.
>
> The scaffold splits for the datasets on moleculenet.org have been used only for classification tasks and not regression. Moreover, this is not used in the papers reported in paperwithcode for the datasets we use. We want to compare our results to them and we use the same procedure and test sets as the papers reported in paperwithcode.
>
> For the question of hyperparameters, we had to make compromises because of the page limit and we don’t report all our experiments that helped choosing the hyperparameters (see answer to reviewer #1 for an example where we tested many different GNN architectures but kept only the best, GAT). For the training details, we describe them in the part where we describe our model, what other training details do you want us to report? We will be happy to add them to the paper.
>
> For the most peripheral node, we explain in section 3.1, at the end if it, you may have missed it.

---

### Official Review · Reviewer_xZRb · 2024-10-08

**Confidence:** 4

**Summary:**

The paper proposes TChemGNN, a GNN to predict molecular properties from their graph representation. TChemGNN integrates local and global 3D molecular features into the node features. It is evaluated on the ESOL and FreeSolv benchmark, where it outperforms existing models.

**Strengths:**

The only part that I find innovative and surprising is the fact that no pooling is performed, but rather only the feature of a single node provides the signal for the whole graph. However, this part is not investigated at all, which in my opinion is a shame.

**Weaknesses:**

I'm confused with the claims made in this paper. First, they integrate 3D and molecular features, as it was done years ago in models like SchNet or DimeNet(++). Therefore, the paper does not present anything "innovative", as the authors claim in the abstract. Second, the title of this paper hints at its efficiency, which is however never evaluated in the paper. The authors never clarify _in what respect_ this model is more efficient. For example, GAT is among the slower graph convolutions, replacing it with GIN should make this architecture even more efficient. The model has only 13k parameters, but there is no chance to know what would happen with more parameters. In summary, the innovation and efficiency claims are unsubstantiated and the experimental part is lacking.

**Justification:**

I suggest to resubmit this paper when it's ready and when its claims are better substantiated. It would also be interesting to expand on the "no-pooling" finding, which in my opinion is the most interesting insight in this paper.

---

> ### Author Rebuttal · Authors · 2024-10-19
>
> Dear reviewers and AC,
> First we want to thank the reviewers for their feedback on our submission, which we find very valuable. We are happy to get such good quality reviews, even if they lean toward rejection.
>
> It seems to us that all the reviewers agree to reject the paper and it is going to be difficult to change your decisions. But we take it as an interesting challenge and we will try to do our best to convince you about the interest of the paper for the community.
> Since the reviewers have similar comments, we have decided to write a global rebuttal to all reviewers that we include in our answer to reviewer #1. In addition, we will make short rebuttals to the individual reviewers to address particular comments.
>
> Datasets: we evaluate our model on 2 datasets. We fully agree, it is quite small. However, and we hope you agree with us, in the field of graph machine learning, many papers are based on the scores on Cora, Citeseer and Pubmed which are very limited datasets, but despite that, the field has developed well and it is possible to show important results on a small and limited number of datasets. We agree it would be better and more convincing with more, richer datasets but there are not so many. Secondly, there exists several other molecule datasets but we want to focus on regression tasks at the molecule level, not classification so it reduces our choices (we are working with chemists on this regression task on a private dataset and we choose to focus on regression in the paper) We will add a sentence to clarify that in the manuscript. Third, we have chosen 2 emblematic datasets, well known, representative and where there is a leaderboard score on them on the “paperwithcode” webpage. So it is easy for everyone to check and track the scores on these datasets. We hope we can convince you it is fair, we have 2 datasets but they are representative and the task is not trivial.
>
> The model: A first look at the model may give the impression we are quite behind the state of the art, with nothing really new. It is “just” 4 layers of GAT. There are a few innovations (3D properties as node features, no pooling), but we agree, it is not groundbreaking. The trend now is to build foundation models. We are going against the trend but we are happy to do so. The main idea of the paper is to show that if you provide a small GNN with global molecule information (3D) at each node, it is able to combine it efficiently with local features for the prediction. It is not really about creating a new GNN. This may not be clear enough in the paper and we will modify the text to make it clearer. We gave a name to our network just so that it can be referred to nicely on benchmarks and hopefully paperwithcode, but it may give the false impression that we claim the main contribution is a new GNN. We insist that the important information in the paper is more conceptual, where we help the GNN to access global information, and simply providing this information at the node level seems to be efficient. It could help in the future building more efficient GNNs, that is what we hope and why we want to share this result with the community.
>
> The results and comparing with other models: We took the paperwithcode website as the target for our evaluations, because it is totally open, with an updated leaderboard of models. We show that our approach is better than bigger models such as Uni-mol (foundation model) for the 2 datasets. We report our results on the same test sets as the one mentioned in paperwithcode or, if not, in the best model paper. We think using these 2 datasets gives enough evidence to show the importance of adding global 3D features to a small GNN, at least for these regression tasks. What is interesting is that bigger models are supposed to be powerful enough to learn to extract global properties (and 3D information) by themselves for their tasks. However it does not seem to be the case since we get better scores. We are considering adding this point in the discussion of our paper. Bigger models may be somehow still limited in their ability to grasp large scale information on a graph and make an efficient combination with local properties.
>
> We also want to point out that we are collaborating with chemists in the project and this paper is the result of this collaboration. That is why we spend time describing the neural net, and why we choose to go for a “simple” neural architecture. The figure describing the model is not really needed for ML researchers but it may help Chemists understand it. We have the hope that it will be used in chemistry, not just on the ML benchmark datasets. That also explains why we make use of this hybrid approach of a neural net together with chemistry features (which is against the trend in ML). But we think the results we got are also of interest for the ML community (global + local information combined and the limits of present NNs to make this combination), that is why we submit in a ML venue, NLDL.
>
> Particular answer to Reviewer #1:
>
> Thank you for your suggestions. We have tried several other GNN models (without finetuning them too much though) but we don’t report it in the paper to not overwhelm the reader with information and focus on the 3D-local feature rather than the possible architectures. We report the RMSE of some models we used:
> GINConv (RMSE: 0.8338),
> MFConv (RMSE: 0.6915),
> SGConv (RMSE: 0.7059),
> GATv2Conv (RMSE: 0.7838),
> GravNetConv (RMSE: 0.7461),
> GraphConv (RMSE: 0.8422),
> SAGEConv (RMSE: 0.8638),
> GCNConv (RMSE=0.9402),
> HypergraphConv (RMSE: 1.1994),
> SuperGATConv (RMSE: 0.9114),
> EGConv (RMSE: 1.2607),
> AntiSymmetricConv (RMSE: 1.1491)  the nearest one for this type of prediction) or applied models for large datasets, the results would not even meet the benchmark scores.
> For example, for GIN RMSE = 0.8338 (it’s even lower than the SOTA model).

---

### Meta-Review · Area_Chair_7jcw · 2024-11-02

**Recommendation:** Reject
**Confidence:** 4

**Metareview:**

The paper proposes TChemGNN, a graph neural network (GNN) model for predicting molecular properties by integrating both local and global 3D molecular features.
The suggested approach outperforms the existing models on the ESOL and FreeSolv datasets, with the main innovation highlighted as a no-pooling design where a single node's features provide the graph's signal. However, the reviewers raise several concerns that limit the paper's suitability for acceptance in its current form.

Strengths:
1) Performance: All reviewers noted that TChemGNN demonstrates strong results on the chosen benchmarks, with effective integration of local and global 3D molecular features.
2) No-Pooling Design: The no-pooling approach, where a single node readout is used instead of pooling, was seen as innovative and promising by multiple reviewers.

Weaknesses:
1) Reviewers pointed out that integrating 3D features is not novel, with similar techniques seen in models like SchNet and DimeNet++. Consequently, they suggested that the authors should moderate their claims of innovation on this part.
2) The paper's reliance on only two datasets (ESOL and FreeSolv) was considered insufficient to support generalizability. Reviewers recommended including more datasets from the Open Graph Benchmark (OGB) or others used in molecular graph property prediction.
3) Reviewers noted that claims about efficiency and novelty were not sufficiently backed by evidence. In particular, comparisons with other GNN architectures lacked tuning, making it difficult to establish the model's performance and efficiency relative to the state-of-the-art.
4) While the authors aimed to make the paper accessible to non-AI researchers (e.g., chemists), the reviewers felt that a machine learning venue like NLDL requires a more targeted AI-focused style, with greater emphasis on methodological rigor.

In sum, reviewers ( with a high confidence average of 3.66)  suggest resubmission with more robust baselines, error bounds, and detailed hyperparameter choices. They also recommend further validation of efficiency claims and discussion of alternative molecular representations.
In reviewing the paper and the reviewer's feedback,  I see promise in this work, particularly in the innovative no-pooling design. However, as all reviewers highlighted, the paper would benefit from stronger evidence, more robust comparisons, and clearer contextualization within the current ML literature. Since the authors did not submit any revisions, and no reviewers support the work for even weak acceptance, the paper may not be ready for acceptance in its current format.

**Suggested Changes To The Recommendation:**

3: I agree that the recommendation could be moved up

---

### Decision · Program_Chairs · 2024-11-06

Reject